# ON SARCASM DETECTION WITH OPENAI GPT-BASED MODELS

## ABSTRACT

Sarcasm is a form of irony that requires readers or listeners to interpret its intended meaning by considering context and social cues. Machine learning classification models have long had difficulty detecting sarcasm due to its social complexity and contradictory nature.

This paper explores the applications of the Generative Pretrained Transformer (GPT) models, including GPT-3, InstructGPT, GPT-3.5, and GPT-4, in detecting sarcasm in natural language. It assesses the differences in sarcasm detection between GPT models with and without domain context, and tests fine-tuned and zero-shot models of different sizes.

The GPT models were tested on the political and balanced (pol-bal) portion of the popular Self-Annotated Reddit Corpus (SARC 2.0) sarcasm dataset. In the fine-tuning case, the largest fine-tuned GPT-3 model achieves accuracy and $F_1$-score of 0.81, outperforming prior models. In the zero-shot case, the latest GPT-4 model yields an accuracy of 0.71 and $F_1$-score of 0.75. Other models score lower. Moreover, domain context does not enhance fine-tuning and reduce zero-shot performance. Additionally, a model's performance may improve or deteriorate with each release, highlighting the need to reassess performance after each release.

## 1 INTRODUCTION

Both humans and artificial intelligence have difficulty interpreting sarcasm correctly. It is especially challenging for textual inputs where body language and speaker intonation are absent. Sarcasm-detecting agents (i.e., systems that detect sarcasm in texts) are tested on their ability to interpret context when determining whether a textual statement is sarcastic (Kruger et al., 2005).

The development of an accurate sarcasm detection systems holds great potential for improving human-computer interactions, given that sarcasm is widely used in human conversations (Olkoniemi et al., 2016). To detect sarcasm in text-based social interactions, a model with contextual knowledge and social understanding capabilities is needed. There has been extensive work in detecting sarcasm, especially with the Self-Annotated Reddit Corpus (SARC 2.0) dataset (Khodak et al., 2017). Sarcasm detection models with the highest performance rely on Transformers (Potamias et al., 2020; Sharma et al., 2023; Li et al., 2021), recurrent neural networks (Kumar et al., 2020; Potamias et al., 2020; Ilić et al., 2018), and/or feature engineering (Hazarika et al., 2018; Li et al., 2021).

OpenAI Generative Pre-trained Transformer (GPT) models have shown effectiveness in natural language understanding tasks (Chen et al., 2023a). However, to our knowledge, there is no comparative study of fine-tuning and zero-shot methods for sarcasm detection, no exploration of how domain context affects said models, and no study of differently versioned ChatGPT models' sarcasm detection abilities.

This paper aims to fill this gap by analyzing the performance of GPT models in detecting sarcasm using the SARC 2.0 political, balanced (pol-bal) dataset. For the sake of brevity, from hereon, we will refer to this dataset as *pol-bal* dataset. Our research questions (RQs) are as follows.

**RQ1.** How does model size affect the ability of fine-tuned GPT-3 models to detect sarcasm?

**RQ2.** What are the characteristics of the top-performing zero-shot GPT model?

**RQ3.** How does domain context affect a GPT model's sarcasm detection performance?

**RQ4.** How is zero-shot learning affected by different versions of the same GPT model?

Our key **contribution** is analyzing how different model sizes, versions, learning methods, and domain context knowledge influence GPT models' ability to detect sarcasm.

The rest of this article is structured as follows. Section 2 presents a literature review. Section 3 covers the methodology of our experiments, while Section 4 discusses the results. Finally, Section 5 concludes the paper.

## 2 LITERATURE REVIEW

The following literature review aims to provide an overview of popular sarcasm detection datasets, research on the *pol-bal* dataset, implications of Large Language Models (LLMs) on sarcasm detection, and GPT models' abilities for other natural language understanding (NLU) tasks.

### 2.1 SARCSAM DETECTION DATASETS

Multimodal Sarcam Detection Dataset (MUStARD), developed by Castro et al. (2019), consists of 690 (50% sarcastic) scenes from four television shows with situational context given in the form of respective speakers and previous dialogue . It intends to challenge multi-modal models with sarcasm detection given situational context. However, the dataset may also be used for text-based classification.

The iSarcasmEval dataset (Abu Farha et al., 2022) contains 6,135 (21% sarcastic) English tweets from English speakers asked to provide their own sarcastic and non-sarcastic tweets. It also provides self-annotated Arabic sarcastic statements. This SemEval task consists of three subtasks: sarcasm detection, sarcasm categorization, and pairwise sarcasm identification.

Providing 362 (50% sarcastic) statements, SNARKS (Ng et al., 2022) uses a contrastive minimal-edit distance (MiCE) setup, presenting a binary choice sarcasm detection task. SNARKS omits sarcastic statements requiring factoid-level knowledge as well as the comment thread leading to a sarcastic statement. SNARKS's sarcasm detection challenge with a MiCE setup tests a model's understanding of a word's literal meaning and its knowledge of its features without considering its context.

#### 2.1.1 DATASET UNDER STUDY: SARC 2.0 *pol-bal*

SARC is a large dataset containing 1.3 million sarcastic Reddit comments from various sub-reddits. A popular benchmark is a subset of this dataset that contains a balanced sample of sarcastic and non-sarcastic comments from the r/politics sub-reddit. This subset is denoted *pol-bal*; it poses the challenge for a model to understand sarcasm within a real social interaction—with situational context—while considering background factoid-based political knowledge. Evaluation on *pol-bal* dataset offers insight into a model's ability to understand sarcasm in a real-world scenario. When detecting sarcasm, especially when a particular political background is required, the researcher's model must be robust.

An interesting feature of the *pol-bal* dataset is its balancing method, where each observation includes a comment thread and two replies to said thread: one sarcastic and one non-sarcastic. Thus, both train and test data are ideally balanced, with 50% of observations being sarcastic and 50% not sarcastic. The *pol-bal* dataset contains 13,668 training and 3,406 testing observations. Each sarcastic comment in the dataset is given one to six preceding comments in its respective thread from the forum.

We have chosen to utilize *pol-bal* dataset due to its unique challenges, high amount of observations, and popularity.

Table 1: Performance of models on SARC *pol-bal* dataset. A – indicates cases in which $F_1$ are not reported.

| Reference | Model | Acc | $F_1$ |
|---|---|---|---|
| Pelser & Murrell (2019) | dweNet | 0.69 | 0.69 |
| Hazarika et al. (2018) | CASCADE | 0.74 | 0.75 |
| Khodak et al. (2017) | Bag-of-Bigrams | 0.77 | – |
| Potamias et al. (2020) | RCNN-RoBERTa | 0.79 | 0.78 |
| Ilić et al. (2018) | ELMo-BiLSTM | 0.79 | – |
| Khodak et al. (2017) | Human (Average) | 0.83 | – |
| Khodak et al. (2017) | Human (Majority) | 0.85 | – |

## 2.2 RESEARCH ON *pol-bal* DATASET

The following models have been constructed to detect sarcasm in SARC *pol-bal* dataset[1]. Summary of the models' performance is given in Table 1.

Khodak et al. (2017) classify their *pol-bal* test set using several models. The best result uses a Bag-of-Bigrams approach and achieves accuracy $\approx 0.77$.

It is not trivial to detect sarcasm, as stated above. According to (Khodak et al., 2017), five human "labelers" attain an average accuracy $\approx 0.83$. A majority vote among the "labelers" improves accuracy to $\approx 0.85$.

Hazarika et al. (2018) develop the ContextuAl SarCasm DEtector (CASCADE) model which uses both content and context modeling to classify an r/politics post's reply as sarcastic. They use a combination of convolutional neural and feature embedding techniques to model the data. This method reaches an accuracy $\approx 0.74$ and $F_1 \approx 0.75$.

Pelser & Murrell (2019) use a dense and deeply connected model in an attempt to extract low-level features from a sarcastic comment without the inclusion of its respective situational context. This method manages to attain accuracy $\approx 0.69$ and $F_1 \approx 0.69$.

Ilić et al. (2018) attempt to classify sarcasm based on morpho-syntactic features of a sarcastic statement. This method uses ELMo word embeddings passed through a BiLSTM to classify sarcasm. It achieves accuracy $\approx 0.79$.

Potamias et al. (2020) propose a method called RCNN RoBERTa, which uses RoBERTa embeddings fed into a recurrent convolutional neural network to detect sarcasm. This method achieves an accuracy $\approx 0.79$ and $F_1 \approx 0.78$.

Thus, two methods achieve state-of-the-art accuracy $\approx$ (Ilić et al., 2018; Potamias et al., 2020).

## 2.3 RESEARCH ON SARCASM DETECTION WITH LLMS

Srivastava et al. (2022) create the BIG-bench benchmark for natural language understanding tasks with large language models. It details the usage of eight differently-sized OpenAI GPT-3 models from (Brown et al., 2020) on 204 natural language tasks in a zero-shot and few-shot manner. In particular, they conduct $[0, 1, 2, 5]$-shot testing on GPT-3 with observations from the SNARKS sarcasm

---

[1]Li et al. (2021) propose a method for detecting sarcasm by addressing the notion that sarcasm detection is based on common sense. Their method is based on using BERT- and COMET-generated related knowledge. This method yields accuracy $\approx 0.76$ and $F_1 \approx 0.76$. They are using another political subset of SARC 2.0. Thus, we cannot compare their performance with that of other papers.

Choi et al. (2023) create a benchmark for social understanding for LLMs; they group social understanding tasks into five categories, including humour and sarcasm. The SARC 2.0 dataset is included in theirs. A DeBERTa-V3 model is found to be the best at detecting sarcasm among the other LLMs tested. However, they do not use the *pol-bal* subset, so the results are not comparable.

Sharma et al. (2023) leverage BERT and fuzzy logic to detect sarcasm. Despite the fact that they present performance results for SARC 2.0, the data subset in their study mixed pol and main SARC 2.0 data subsets, making it impossible to compare their results to other papers in this section.

dataset (discussed in Section 2.1). As they are using a different dataset, their work is complementary to ours.

Mu et al. (2023) compare the sarcasm-detecting abilities of GPT-3.5-turbo, OpenAssistant, and BERT-large on the iSarcasmEval dataset (discussed in Section 2.1). They find that BERT-large is the most performative on this dataset. This work is complementary to ours as it uses a different dataset and a single GPT model.

Choi et al. (2023) create a benchmark for social understanding for LLMs; they group social understanding tasks into five categories, including humor and sarcasm, which includes the SARC 2.0 dataset. As discussed in Section 2.2, their work is complementary to ours.

Băroiu & Trăuşan-Matu (2023) use a fine-tuned GPT-3 curie and zero-shot text-davinci-003 models on the MUStARD dataset (discussed in Section 2.1) and achieve top $F_1 = 0.77$. This work is complementary to ours as they are using a different dataset.

### 2.4 RESEARCH ON OTHER NATURAL LANGUAGE UNDERSTANDING TASKS USING GPT MODELS

Chen et al. (2023b) compare $[0, 1, 3, 5]$-shot LaMDA-PT, $[0, 2, 3, 4, 10, 15, 16]$-shot FLAN, and fine-tuned popular approaches to $[0, 3, 6, 9]$-shot InstructGPT models (text-davinci-001, text-davinci-002) and the text-davinci-003 GPT-3.5 model on 21 datasets across 9 different NLU tasks not including sarcasm detection. They find that the GPT-3.5 model performs better than the other models in certain tasks like machine reading comprehension and natural language inference, but performs worse than other models in sentiment analysis and relation extraction.

## 3 METHODOLOGY

Our dataset under study—*pol-bal*—is discussed in Section 2.1.1. Using this dataset, we study twelve versions of GPT models shown in Table 2. OpenAI GPT are generative models pre-trained on a large corpus of text data (Brown et al., 2020). GPT models are pre-trained to predict the next token in a given document, learning to estimate the conditional probability distribution over its vocabulary given the context; see Zhao et al. (2023) for review. With pre-training, GPT models are equipped with a vast amount of language knowledge and world information, which, in conjunction with their large parameter count, allows them to excel at natural language tasks (Brown et al., 2020).

The four model families tested in this paper are GPT-3, InstructGPT, GPT-3.5, and GPT-4. Technical details of the models are given in Brown et al. (2020); Ouyang et al. (2022); OpenAI (2022; 2023a). In short, InstructGPT models are fine-tuned GPT-3 models aligned with human intent (Ouyang et al., 2022). The GPT-3.5 models, like text-davinci-003, are based on code-based GPT-3 models (Zhao et al., 2023). GPT-3.5-turbo is a version of GPT-3.5 trained to follow instructions and give detailed responses; it is a ChatGPT model (OpenAI, 2022). GPT-4 is currently the most successful GPT model for professional and academic tasks; it is also a ChatGPT model (OpenAI, 2023a).

To answer this work's research questions, we initially developed prompts which wrap each observation from the dataset. Subsequently, we fine-tuned and tested the base GPT-3 models, zero-shot tested the GPT-3, InstructGPT, GPT-3.5, and GPT-4 models, and finally performed statistical analyses on their results as discussed below.

### 3.1 PROMPT DEVELOPMENT STAGE

This sarcasm classification problem was addressed by creating three prompts: one with domain context, one without domain context, and a system prompt for zero-shot testing (shown in Figure 1). The domain context prompts were first developed for fine-tuned models—hence the usage of the recommended[2] prompt/completion delimiter \n\n###\n\n, then applied to the zero-shot models for consistency. During fine-tuning and at inference, the `<thread>` placeholder was replaced with a given observation's context (the thread of text leading up to the response), while the `<reply>` placeholder was replaced with an observation's response.

---

[2] As per the recommendations of (OpenAI, 2023b).

Table 2: GPT models under study. * denotes ChatGPT model, $\sim$ denotes similar order of magnitude. The size of the ada, babbage, curie, and davinci models is reported by Brown et al. (2020). The size of the remaining models is unknown. However, it is conjectured that the size of InstructGPT and GPT-3.5 models is similar in magnitude to their predecessors (Zhao et al., 2023), while the size of GPT-4 is larger than of their predecessors (Schreiner, 2023).

| Model Family | Version | # of Parameters | Release Date |
|---|---|---|---|
| GPT-3 | ada | 0.4 B | June 2020 |
| GPT-3 | babbage | 1.3 B | June 2020 |
| GPT-3 | curie | 6.7 B | June 2020 |
| GPT-3 | davinci | 175.0 B | June 2020 |
| InstructGPT | text-ada-001 | $\sim 0.4$ B | January 2022 |
| InstructGPT | text-babbage-001 | $\sim 1.3$ B | January 2022 |
| InstructGPT | text-curie-001 | $\sim 6.7$ B | January 2022 |
| GPT-3.5 | text-davinci-003 | $\sim 175$ B | January 2022 |
| GPT-3.5 | GPT-3.5-turbo-0301* | $\sim 175$ B | March 2023 |
| GPT.3.5 | GPT-3.5-turbo-0613* | $\sim 175$ B | June 2023 |
| GPT-4 | GPT-4-0314* | $\geq 175$ B | March 2023 |
| GPT-4 | GPT-4-0613* | $\geq 175$ B | June 2023 |

```
Does the reply to
this thread in
Reddit r/politics
subreddit seem
sarcastic?\nThread:
\n<thread>\nReply:
\n<reply>\n\n###\n\n
```
(a) Prompt with domain context.

```
Thread:
\n<thread>\nReply:
\n<reply>\n\n###\n\n
```
(b) Prompt w/o domain context.

```
Classify each comment
thread's response as
sarcastic with yes or
no. \n
```
(c) Zero-shot prompt's prefix.

Figure 1: Prompts used for fine-tuning and zero-shot GPT models with and without domain context. `<reply>` and `<thread>` placeholders represent template values that need to be replaced with actual content.

We introduced the `->` delimiter between comments in cases where multiple comments were provided within a given observation (as discussed in Section 2.1.1). An example of such an observation is given in Figure 2.

## 3.2 FINE-TUNING STAGE

To test how the model size and domain context affect a fine-tuned GPT model's ability to detect sarcasm, fine-tuned versions of the GPT-3 ada, babbage, curie, and davinci models were created and tested.

When the experiments were conducted, OpenAI did not allow fine-tuning of GPT-3.5 and GPT-4 models. OpenAI has enabled fine-tuning of GPT-3.5 models as of August 22, 2023, and will enable

```
Classify each comment thread's response as sarcastic with yes or no.
\n Does the reply to this thread in Reddit r/politics subreddit seem
sarcastic?\nThread: \nJust finished watching the debate. I love the
President! -> Agreed! Can't wait for the next event!\nReply: \nOh, the
suspense is killing me!\n\n###\n\n
```

Figure 2: A zero-shot test prompt with domain context and two comments in a thread.

fine-tuning of GPT-4 models later in the year (Peng et al., 2023). Due to time constraints, we could not fine-tune GPT-3.5 and GPT-4 models, but we plan to do so in the future.

The fine-tuning was performed using the following hyperparameter values: a learning rate of 0.1, 4 epochs, a prompt loss weight of 0.01, and a batch size of 16. Two variants of each GPT-3 model were created: one trained on the domain context prompt and one trained on the no domain context prompt shown in Figures 1a and 1b, respectively.

Each fine-tuned model was trained on the *pol-bal* train set, with its desired completion being " yes" or " no". Once fine-tuned, a model is tested on the *pol-bal* test set with its temperature set to zero (in order to reduce the variability of the model's output) and restricted to outputting one token.

### 3.3 ZERO-SHOT STAGE

Zero-shot testing methods were used to explore further how domain context, as well as different generations and re-releases, impact a GPT model's sarcasm detection performance. Each of the 12 models (shown in Table 2) was tested with and without domain context using the corresponding prompt.

The prompts in Figures 1a and 1b were prefixed with the sentence in Figure 1c during inference on zero-shot models—and added as a system prompt for ChatGPT models. Thus, the final prompt asks the model to answer "yes" or "no". In practice, the models return complete sentences containing "yes" and "no" words. Model output is mapped to a sarcastic or not sarcastic label using a regular expression that checks for "yes" or "no".

Some models may also return empty outputs or corrupted sentences without "yes" or "no" keywords (or their synonyms). The logit bias was introduced to the models to force them to return only "yes" or "no" tokens.

The output of the models with and without logit bias is compared. Also reported are the number of observations that did not return "yes" or "no" keywords.

### 3.4 DATA ANALYSIS STAGE

Accuracy and $F_1$-score metrics are used to measure the performance of classification methods. The metrics we used are the same as those used by other researchers studying this dataset (see Section 2.1.1 for details). From the metrics perspective, the sarcastic label is considered *positive*, whereas the non-sarcastic label is considered *negative*.

To establish a baseline, we can use the naive ZeroR classifier, which labels every test observation as sarcastic. Given that the dataset is balanced, ZeroR classifier accuracy $= 0.5$ and $F_1 \approx 0.67$.

As described in Section3.3, some models cannot label some observations. Such observations are removed from the list of observations used to calculate accuracy and $F_1$-score.

McNemar's test (McNemar, 1947) is used to determine if there was a statistically significant difference between the classification methods' performance, as recommended by Dietterich (1998).

## 4 RESULTS

Below are the results of the experiments described in Section 3. Sections 4.1–4.4 present the results as answers to our four research questions (defined in Section 1). Section 4.5 discusses the results and identifies and identify possible threats to their validity.

### 4.1 RQ1: HOW DOES MODEL SIZE AFFECT THE ABILITY OF FINE-TUNED GPT-3 MODELS TO DETECT SARCASM?

The **answer to RQ1** is as follows. In Table 3, the accuracy and $F_1$-scores of fine-tuned models increased monotonically with model size. Using the smallest model — ada — produces the worst results, while using the largest model — davinci — produces the best results. To the best of our knowledge, a fine-tuned davinci model with and without domain context achieves state-of-the-art

results (accuracy $= 0.81$ and $F_1 = 0.81$) when applied to the *pol-bal* dataset. To ensure that the difference in performance was statistically significant, we performed pairwise analyses using McNemar's test for all fine-tuned models. According to the test, the differences between all the models are statistically significant ($p \leq 0.021$), see Appendix A for details.

## 4.2 RQ2: WHAT ARE THE CHARACTERISTICS OF THE TOP-PERFORMING ZERO-SHOT GPT MODEL?

Table 3 shows that the GPT-3, InstructGPT, and GPT-3.5 text-davinci-003 models perform worse than the ZeroR classifier (described in Section 3.4). Although logit bias reduces the count of missing observations, low accuracy and $F_1$-scores indicate that the models cannot differentiate between sarcastic and non-sarcastic comments. Even for top-performing models (e.g., GPT-4), the addition of logit bias leads to performance degradation.

The GPT-3.5-turbo and GPT-4 models perform better than the ZeroR baseline. However, except for GPT-4 GPT-4-0613, their performance is lower than those of the simpler models shown in Table 1.

GPT-4 GPT-4-0613 won the zero-shot "challenge" with an accuracy $\approx 0.71$ and $F_1 \approx 0.75$. The model answered "yes" or "no" to most (99.79%) of observations. Peculiarly, the prompt without domain context produced better results than the prompt with domain context.

Based on this analysis, the **answer to RQ2** is as follows. In the *pol-bal* dataset, only the most sophisticated GPT model (i.e., GPT-4 GPT-4-0613) can detect sarcasm competitively using the zero-shot approach. Table 1 shows that this model performs poorly, placing it second-last among the prior models.

## 4.3 RQ3: HOW DOES DOMAIN CONTEXT KNOWLEDGE AFFECT A GPT MODEL'S SARCASM DETECTION PERFORMANCE?

The effect of domain context (see Figure 1 for prompts) on each model is shown in Table 3.

The accuracy and $F_1$-scores of fine-tuned models vary slightly ($\pm 0.01$) with and without domain context. McNemar's test shows that this variability is not statistically significant ($p > 0.05$).

There are 11 statistically significant ($p < 0.05$) cases of variability in the outputs of GPT-3, GPT-3.5, and InstructGPT models due to domain context presence or absence for zero-shot models. As discussed in the answer to RQ2, given that the models' performance is below the ZeroR classifier, it is difficult to draw any conclusions from their performance. We conjecture that the random nature of model outputs may explain this variability to some extent.

Domain context reduces the number of missed observations for GPT-3 models (without logit bias) from $\approx 100\%$ to a range between 97.97% and 99.65%. However, this is a very modest improvement. Furthermore, the classification performance for these subsets of observations is poor (accuracy $\leq 0.02$). In other words, domain context can sometimes help elicit the label, but the label is usually incorrect.

According to McNemar's test, both versions of GPT-4 (without logit bias) show statistically different results in the presence and absence of domain context ($p < 10^{-16}$). Surprisingly, both models perform better without domain context!

Thus, the **answer to RQ3** is as follows. In the fine-tuning case, domain context (i.e., the fact that the data are derived from Reddit's r/politics subreddit) is irrelevant. In the zero-shot case, GPT-4 models may be hindered by the domain context. This finding may indicate that a better method of passing domain context to the model is needed.

## 4.4 RQ4: HOW IS ZERO-SHOT LEARNING AFFECTED BY DIFFERENT VERSIONS OF THE SAME GPT MODEL?

In our work, we have two models with two versions, namely

1. GPT-3.5-turbo released in March (0301) and June (0614) of 2023 and
2. GPT-4 released in March (0314) and June (0614) of 2023.

Table 3: Results of classification of the *pol-bal*. We highlight the best results for fine-tuned (Fine-tune = Y) and zero-shot (Fine-tune = N) models in **bold**. A column labeled Missed shows the percentage of observations (out of 3406 test observations) where a model returned no labels. The last column shows McNemar's test $p$-values; a value in a column is *italicized* when there is a statistically significant ($p < 0.05$) difference between the performance of models with and without domain context.

| Model Family | Model | Parameters count | Fine-tune | w/ Bias | w/ domain | | | w/o domain | | | McNemar's $p$-value |
|---|---|---|---|---|---|---|---|---|---|---|---|
| | | | | | Acc | $F_1$ | Missed (%) | Acc | $F_1$ | Missed (%) | |
| GPT-3 | ada | 0.4 B | Y | N | 0.73 | 0.74 | 0.00 | 0.74 | 0.74 | 0.00 | 0.642 |
| GPT-3 | babbage | 1.3 B | Y | N | 0.75 | 0.76 | 0.00 | 0.75 | 0.75 | 0.00 | 0.961 |
| GPT-3 | curie | 6.7B | Y | N | 0.77 | 0.77 | 0.00 | 0.78 | 0.78 | 0.00 | 0.109 |
| GPT-3 | davinci | 175.0 B | Y | N | **0.81** | **0.81** | 0.00 | **0.81** | **0.81** | 0.00 | 0.954 |
| GPT-3 | ada | 0.4 B | N | Y | 0.50 | 0.00 | 0.00% | 0.50 | 0.67 | 0.00% | 1.000 |
| GPT-3 | babbage | 1.3 B | N | Y | 0.50 | 0.01 | 0.00% | 0.50 | 0.00 | 0.00% | 1.000 |
| GPT-3 | curie | 6.7 B | N | Y | 0.50 | 0.66 | 0.00% | 0.50 | 0.00 | 0.00% | 0.959 |
| GPT-3 | davinci | 175.0 B | N | Y | 0.49 | 0.05 | 0.00% | 0.51 | 0.63 | 0.00% | 0.081 |
| GPT-3 | ada | 0.4 B | N | N | 0.36 | 0.04 | 97.97% | – | – | 0.00% | *0.000* |
| GPT-3 | babbage | 1.3 B | N | N | 0.58 | 0.44 | 99.65% | – | – | 0.00% | *0.008* |
| GPT-3 | curie | 6.7 B | N | N | 0.28 | 0.10 | 99.27% | – | – | 0.00% | *0.000* |
| GPT-3 | davinci | 175.0 B | N | N | 0.33 | 0.45 | 99.47% | – | – | 99.97% | *0.000* |
| InstructGPT | text-ada-001 | $\sim$ 0.4 B | N | Y | 0.51 | 0.53 | 0.00% | 0.50 | 0.66 | 0.00% | 0.176 |
| InstructGPT | text-babbage-001 | $\sim$ 1.3 B | N | Y | 0.49 | 0.23 | 10.66% | 0.48 | 0.63 | 0.00% | 0.290 |
| InstructGPT | text-curie-001 | $\sim$ 6.7 B | N | Y | 0.45 | 0.36 | 0.00% | 0.50 | 0.67 | 0.00% | *0.001* |
| InstructGPT | text-ada-001 | $\sim$ 0.4 B | N | N | 0.52 | 0.43 | 0.47% | 0.53 | 0.61 | 75.63% | *0.000* |
| InstructGPT | text-babbage-001 | $\sim$ 1.3 B | N | N | 0.50 | 0.03 | 0.94% | 0.48 | 0.64 | 4.79% | 0.157 |
| InstructGPT | text-curie-001 | $\sim$ 6.7 B | N | N | 0.49 | 0.65 | 0.18% | 0.50 | 0.67 | 0.09% | *0.001* |
| GPT-3.5 | text-davinci-003 | $\sim$ 175.0 B | N | Y | 0.52 | 0.67 | 0.00% | 0.54 | 0.68 | 0.00% | 0.239 |
| GPT-3.5 | GPT-3.5-turbo-0301 | $\geq$ 175.0 B | N | Y | 0.50 | 0.67 | 0.00% | 0.50 | 0.67 | 0.00% | *0.000* |
| GPT-3.5 | GPT-3.5-turbo-0613 | $\geq$ 175.0 B | N | Y | 0.50 | 0.44 | 0.00% | 0.49 | 0.42 | 0.00% | 1.000 |
| GPT-3.5 | text-davinci-003 | $\sim$ 175.0 B | N | N | 0.55 | 0.68 | 0.06% | 0.59 | 0.64 | 2.58% | *0.001* |
| GPT-3.5 | GPT-3.5-turbo-0301 | $\geq$ 175.0 B | N | N | 0.53 | 0.29 | 0.00% | 0.50 | 0.00 | 0.00% | *0.004* |
| GPT-3.5 | GPT-3.5-turbo-0613 | $\geq$ 175.0 B | N | N | 0.50 | 0.39 | 13.07% | 0.47 | 0.30 | 0.00% | *0.000* |
| GPT-4 | GPT-4-0314 | $\geq$ 175.0 B | N | Y | 0.50 | 0.67 | 0.00% | 0.50 | 0.67 | 0.00% | 0.219 |
| GPT-4 | GPT-4-0613 | $\geq$ 175.0 B | N | Y | 0.58 | 0.70 | 0.00% | 0.63 | 0.72 | 0.00% | *0.000* |
| GPT-4 | GPT-4-0314 | $\geq$ 175.0 B | N | N | 0.58 | 0.70 | 0.00% | 0.61 | 0.71 | 0.06% | *0.000* |
| GPT-4 | GPT-4-0613 | $\geq$ 175.0 B | N | N | **0.68** | **0.75** | 0.00% | **0.71** | **0.75** | 0.21% | *0.000* |

McNemar's test $p$-values for these models are provided in Appendix B. According to the results, the difference between releases is statistically significant ($p < 0.00032$) when domain context is present or absent. There is only one exception to this rule: GPT-3.5-turbo with domain context.

As shown in Table 3, the latest version of GPT-4 performed better than the earlier version (e.g., in the no-domain context case, accuracy improved from 0.61 to 0.71). However, the GPT-3.5-turbo model's performance has declined from release to release (e.g., accuracy decreased from 0.50 to 0.47 in the no-domain context case).

In other words, the answer to RQ4 is as follows. The GPT model's ability to detect sarcasm may decline or improve with new releases, as was observed for other tasks by Chen et al. (2023a).

## 4.5 DISCUSSION AND THREATS TO VALIDITY

This article groups threats to validity into four categories (internal, construct, conclusion, and external) as presented in Yin (2017).

**Internal validity** In order to reduce the risk of errors, all experiments and data analysis were automated using Python scripts (which were cross-reviewed and validated by the authors).

**Construct validity** This study uses accuracy and $F_1$-score to measure models' performance. Previous authors studying this dataset used the same metrics (see Section 2.1.1 for details).

In this study, several formulations of prompts were tested; the most effective ones were reported. Our prompts may be suboptimal, and we encourage the community to create better ones. The performance of the models may be further enhanced with such better prompts.

Furthermore, our research scope did not cover sophisticated prompt engineering techniques, such as $k$-shot (Brown et al., 2020), chain of thought with reasoning (Kojima et al., 2022), or retrieval augmented generation (Lewis et al., 2020) that could improve the models' performance. However,

this study aims not to achieve the best results for sarcasm detection but to examine how various attributes of models and prompts affect classification performance.

Another threat is that we use a regular expression to map the output of models to a label without looking at the output context. Thus, during zero-shot tests, regular expressions may misclassify specific outputs. We have sampled and eyeballed many outputs to mitigate this risk. We observed that the keyword "yes" mapped to "sarcastic" and the keyword "no" mapped to "non-sarcastic" in all cases. Furthermore, if neither label "yes" nor "no" was present, the output was meaningless and could not be categorized.

As mentioned in Section 3.2, fine-tuning of GPT-4 models is not available at this time. We plan to fine-tune GPT-4 models when they become available in the future. Fine-tuned GPT-4 models will likely perform better than the GPT-3 davinci model (if the trend of increasing performance with increasing parameter count discussed in Section 4.1 holds).

**Conclusion validity**   We do not know the exact content of the datasets used to train GPT models (Brown et al., 2020). Therefore, it is possible that GPT models saw *pol-bal* dataset during training. The fact that fine-tuning improves the results significantly over zero-shot experiments may suggest otherwise. At the very least, it may suggest that the models have "forgotten" this information to a significant extent.

**External validity**   Our study is based on a single dataset. Despite its popularity in sarcasm detection studies, our findings cannot be generalized to other datasets. However, the same empirical examination can also be applied to other datasets with well-designed and controlled experiments. Furthermore, this research serves as a case study that can help the community understand sarcasm detection trends and patterns and develop better LLMs to detect complex sentiments, such as sarcasm.

Although the GPT-3 davinci fine-tuned model produced best-in-class prediction results, it does not imply that it will be used for those purposes in practice due to economic concerns. Due to the large number of parameters in LLMs, inference is expensive. The expense may be justified in business cases where sarcasm detection is critical.

## 5   SUMMARY

Humans and machine learning models have difficulty detecting sarcasm. This study examines twelve GPT models' ability to detect sarcasm in the popular *pol-bal* dataset. The results are as follows.

The fine-tuned GPT-3 davinci model with 175 billion parameters outperforms prior models with an accuracy and $F_1$-score of 0.81. In the zero-shot case, only the latest GPT-4 model could detect sarcasm with comparable performance to the prior models (achieving the second-last result). The performance of a model could improve or decline with a new release, which implies that every new release should be reevaluated.

A study was conducted to determine whether domain context could improve models' performance. It was shown that domain context data was irrelevant for fine-tuning and harmful for zero-shot approach. This unintuitive result requires further study.

Practitioners interested in incorporating sarcasm detection into their applications may find this work useful. It may also interest academics who wish to improve machine learning models' ability to detect complex language usage.

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

## A  RESULTS OF THE PAIRWISE MCNEMAR'S TEST FOR FINE-TUNED GPT-3 MODELS

McNemar's test (McNemar, 1947) is applied to predictions of each pair of classification models as per Dietterich (1998) to compare the performance of different-sized fine-tuned GPT-3 models. Figures 3 and 4 show the results of the pairwise comparison of the models trained with and without domain context, respectively. Statistically significant differences are observed in all cases ($p \leq 0.021$).

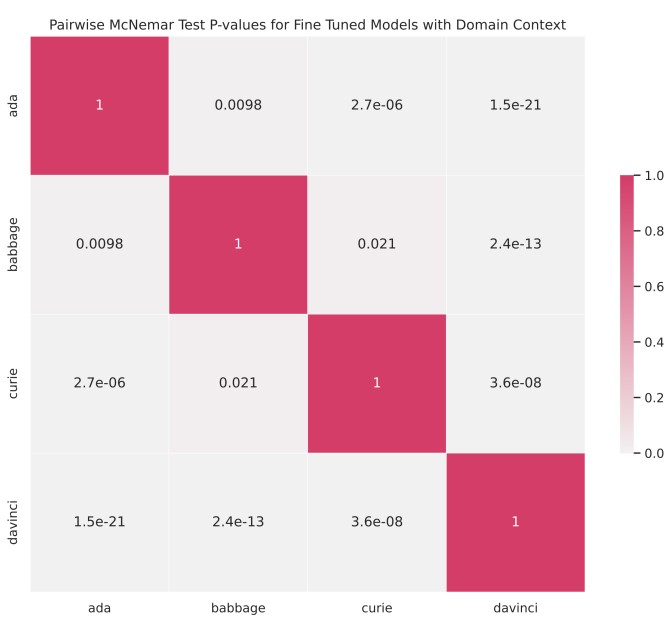

Figure 3: Pairwise McNemar's test of fine-tuned base GPT-3 models with domain context.

## B  RESULTS OF THE PAIRWISE MCNEMAR'S TEST FOR DIFFERENT VERSIONS OF GPT-3.5-TURBO AND GPT-4

In this section, we compare the difference in performance for different versions of GPT-3.5-turbo and GPT-4 models for the zero-shot case (without logit bias). McNemar's test (McNemar, 1947) is applied to predictions of each pair of classification models as per Dietterich (1998). Figures 5 and 6 show the pairwise comparison results of the models tested with and without domain context, respectively. There are statistically significant differences in all cases except for the GPT-3.5-turbo pair of models ($p \approx 0.21$) with the domain context present.

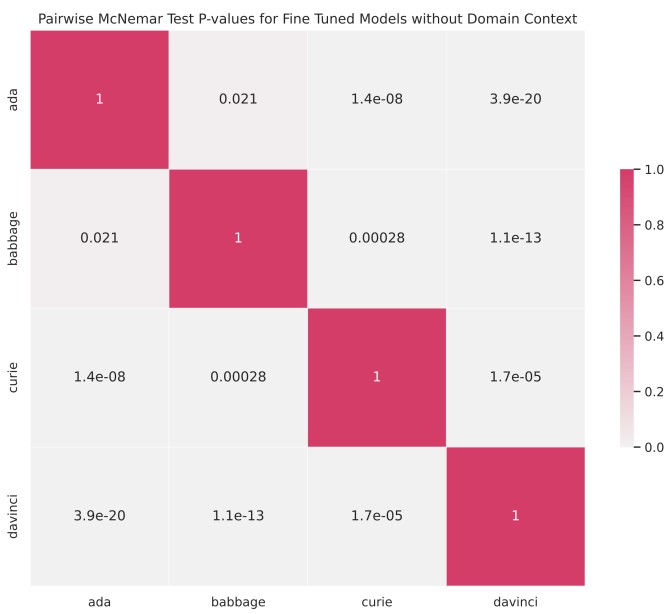

Figure 4: Pairwise McNemar's test of fine-tuned base GPT-3 models without domain context.

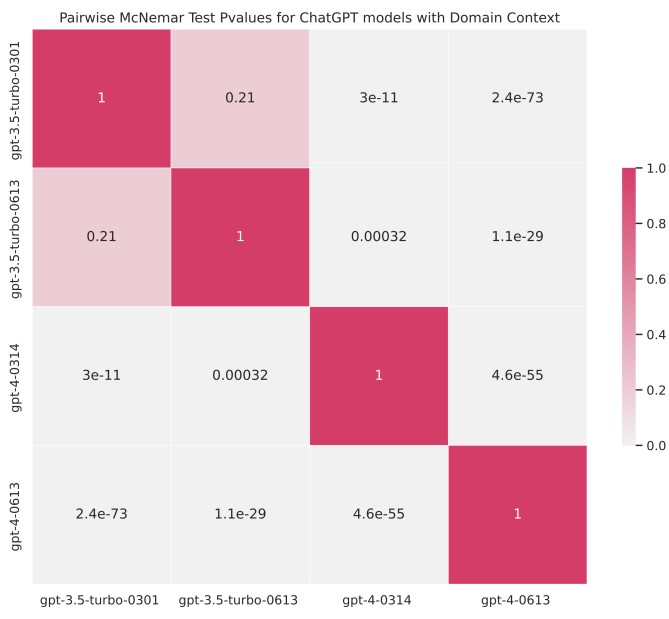

Figure 5: Pairwise McNemar's test of ChatGPT models with domain context.

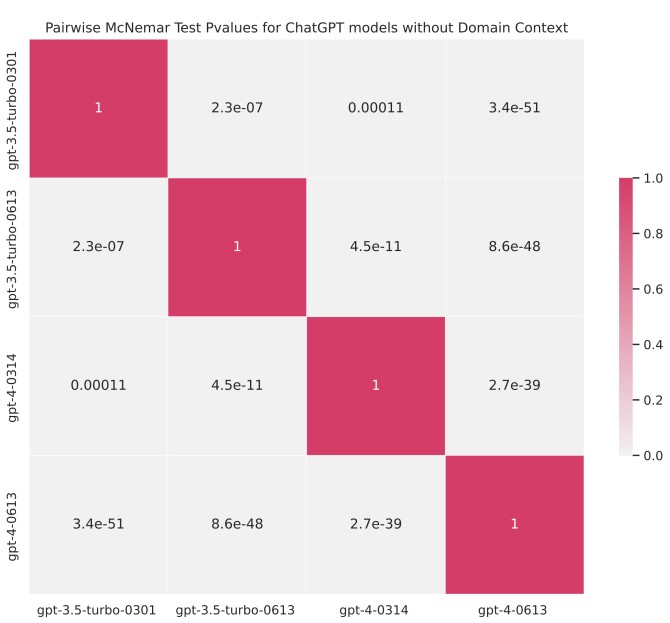

Figure 6: Pairwise McNemar's test of ChatGPT models without domain context.

