# OpenReview forum: "On Sarcasm Detection with OpenAI GPT-based Models"
_ICLR.cc/2024/Conference — Submitted to ICLR 2024_

### Official Review · Reviewer_ir58 · 2023-10-28

**Soundness:** 2 fair
**Presentation:** 2 fair
**Contribution:** 2 fair
**Rating:** 1
**Confidence:** 5

**Summary:**

The main focus of the paper is to analyze the impact of model size, versions, learning methods, and domain context knowledge on GPT models' ability to detect sarcasm. The authors conducted experiments using different versions of GPT models and tested their performance in sarcasm detection using the SARC 2.0 sarcasm dataset from Reddit. The experiments reveal that larger fine-tuned GPT models achieve higher accuracy and F1-scores, and the latest GPT-4 model performs better than earlier versions in zero-shot detection. The paper also highlights the importance of reassessing model performance after each release as the performance of GPT models can vary. Overall, the findings may provide insights into the detection of sarcasm using GPT models.

**Strengths:**

1. The work offers a relatively comprehensive investigation into the utilization of GPT-based LLMs for sarcasm detection.
2. The research showcases experimental outcomes from applying GPT models to the SARC 2.0 sarcasm dataset. Specifically, it demonstrates the proficiency of both fine-tuned and zero-shot GPT variants across multiple model sizes, versions, and context.
3. The conclusions drawn from the research have potential to provide valuable insights and guidance for the direction of sarcasm detection methodologies.

**Weaknesses:**

1. At a holistic level, the manuscript comes across more as an experimental report rather than a meticulously structured technical paper.
2. The choice of prompts is crucial for achieving optimal task performance. However, this paper seems to overlook a comprehensive exploration and display of various text prompt designs. Furthermore, there is an absence of examination into the potential influence of different prompts on the model's performance. This oversight could lead to inherent biases when drawing conclusions.
3. The presentation of experimental results is somewhat lackluster. The paper primarily relies on a single table, devoid of additional results or visual representations, which challenges the reader's confidence in certain analytical outcomes. This limited presentation also means missed opportunities for more intricate and engaging analysis.
4. The experimental scope misses out on comparisons with non-GPT LLMs and also doesn't juxtapose the performance of LLMs against traditional models. Consequently, the findings and conclusions can be perceived as quite constrained.
5. While the paper attempts to conclude its findings, it does not pave the way forward by suggesting potential future work and directions. Offering such insights would be beneficial for subsequent researchers to build upon this study.

**Questions:**

1. You highlight that the most advanced fine-tuned GPT-3 model clocks in with both accuracy and F1-score of 0.81 in sarcasm detection. How does this stack up against prior models? Could you delve deeper and furnish information regarding the performance metrics of preceding models on this identical dataset?

---

> ### Author Response · Authors · 2023-11-21
>
> Thank you for the question! The information is summarized in Table 1 and Section 2.2. Accuracy of the previous five models varies from 0.69 to 0.79; F1 – from 0.69 to 0.78.

---

### Official Review · Reviewer_kn7S · 2023-10-31

**Soundness:** 3 good
**Presentation:** 3 good
**Contribution:** 2 fair
**Rating:** 5
**Confidence:** 4

**Summary:**

Focused on the pol-bal dataset, this study examined the ability of differently-sized and different versions of GPT models to detect sarcasm with and without domain context, fine-tuned and zero-shot.

The authors designed prompts with and without domain context to feed the data to LLMs for fine-tuning and zero-shot tests.
The results led to the answers of the 4 research questions put forward in the introduction, which are:

1. The accuracy and F1-scores of fine-tuned models increases monotonically with model size, and the GPT-3 davinci model achieves the state-of-the-art performance on the pol-bal dataset.
2. In the pol-bal dataset, only the most sophisticated GPT model (i.e., GPT-4 GPT-4-0613) can detect sarcasm competitively using the zero-shot approach.
3. In the fine-tuning case, domain context is irrelevant; In the zero-shot case, GPT-4 models may be hindered by the domain context.
4. The GPT model’s ability to detect sarcasm may decline or improve with new releases.

**Strengths:**

### General

- The paper is well-written and easy to follow.
- The study is the first research of LLMs' performance on sarcasm detection with the pol-bal dataset.
- Analysis of the results is detailed and rigorous.

### RQ1

- The study achieves state-of-the-art performance on the pol-bal dataset with a fine-tuned GPT-3 model.

**Weaknesses:**

### General

- The task of using LLMs to detect sarcasm has been studied before, and has been included in many papers to evaluate the performance of LLMs. As is repeated 4 times in section 2.3, the main distinction from previous works is claimed to be the use of the pol-bal dataset, which don't seem to be a significant contribution.
- The study is based on GPT models from OpenAI only, which causes the results to be biased and not representative of the whole LLMs family.

### RQ2

- The claim in section 4.2 that GPT-3.5-turbo performs better than ZeroR classifier can't be inferred from the experimental results. The accuracies are just around 50% and F1 scores are terribly low.

### RQ3

- The claim in section 4.3 that domain context reduces the number of missed observations for GPT-3 models (without logit bias) from ≈ 100% to a range between 97.97% and 99.65% is not shown in Table 3. Actually the columns of `w/o domain` in these rows are empty, except for the 0.00% missing rate from ada to curie and the 99.97% missing rate for davinci.
- The conclusion about the effect of domain context on classification tasks is valid for the single given form **ONLY**, but other forms of domain context might be more effective. This is mentioned in section 4.5, but I recommend adding other tested domain context forms and results in the appendix.
- The conclusion is insignificant and not decisive.

### RQ4

- Analysis of the performance declination of GPT-3.5-turbo model is doubtful, since the performance is not significantly different from a random classifier.
- The question is not meaningful, because the paper just presents the results and doesn't provide any technical explanation for the performance variation. No insight can be gained from the conclusion, even if a relation between the performance and the corresponding GPT version were found.

**Questions:**

- Is it possible to add more forms of domain context to see how the length, phrasing and other features affect the performance?
- Why are the accuracies without domain context in Table 3 empty for GPT-3 models without logit bias? And why do the missing rates disagree with the claim in section 4.3 about missing rate reduction due to domain context?
- Is it possible to try more LLMs to make the conclusions more representative and general?

---

### Official Review · Reviewer_5eai · 2023-11-03

**Soundness:** 3 good
**Presentation:** 2 fair
**Contribution:** 1 poor
**Rating:** 3
**Confidence:** 2

**Summary:**

This paper presents a set of experiments with a large number of GPT
family of models, investigating their ability to detect sarcasm
with/without domain information provided as part of the prompt in
well as zero-shot setting and through fine-tuning on the task. The
results indicate some trends in the GPT models' success on the task -
the most straightforward one being "larger better". However, the effect of
domain information is unclear.

**Strengths:**

Sarcasm detection, in general, identifying any type of figurative
language is one of the difficult problems in NLP.
The use of context and additional information
(i.e., the domain of the text) is also interesting,
as figurative language requires (situational) information beyond the given text.
The extensive results presented may help practitioners to decide for
the GPT model to use in case the application at hand requires
sensitivity to sarcasm.

**Weaknesses:**

I find the overall information content coming out of the study rather
weak. Knowing the relative success of the GPT models may help some
users of these models. However, the results  are specific to these (rather
opaque) models, and they do not, otherwise, provide further
information on the problem, or the mechanisms/representations useful
for solving it. In this respect, the most interesting research
question is the effect of the domain information. Yet, the experiments
do provide any clear knowledge on this question.
I think the paper would be much more insightful if this question was at the center.
I will list a few concrete points of criticism of the paper below.

- As noted above, I think a more scientifically interesting question is
  the effects/mechanisms at play when using (situational) context to
  detect sarcasm. The results presented are rather unclear on this
  question. A possible reason could be the limited information
  provided (only the fact that the conversation is about politics).
  Presumably the large language models can also infer this from the
  thread provided (e.g., "Just finished watching the debate. I love
  the President" already has a strong indication that the domain is
  politics). Focusing on providing information that is not available
  in the thread, or analyzing the instances where the provided domain
  information gives additional information may have been interesting
  here.

- The use of statistical/quantitative comparisons are good, but there
  are some issues. First, for some of the statistical significance results reported
  multiple comparisons should be taken into consideration (not all
  require compensations, but there are some multiple-comparisons that
  are rather exploratory, and need correction). Second, during
  reporting, expressions like "p ≤ 0.021" are not strictly correct.
  Here, presenting what the p values equals to (up to some significant
  digits), or just indicating that it is less than the level chosen
  would be proper ways of reporting it (I'd personally prefer the
  former).

- The paper would benefit from some revisions that reduced
  "itemize-like" listing into a more coherent narrative (e.g., in
  literature review section). It would also be nicer to the reader to
  use verbal expressions rather than (not-so-standard) notation like
  "accuracy ≈ 0.77". Another minor note: the long footnote (1) should
  probably be part of the text. There are also some minor typographic/language
 mistakes that could be fixed with a revision (e.g., spurious space around punctuation
in the first paragraph of Section 2.1, placement of footnote marks before punctuation,
case (normalization) issues in bibliography...)

**Questions:**

Please see "weaknesses above".

---

### Official Review · Reviewer_XmYa · 2023-11-09

**Soundness:** 2 fair
**Presentation:** 3 good
**Contribution:** 2 fair
**Rating:** 5
**Confidence:** 4

**Summary:**

Inspired by the effectiveness of OpenAI GPT models over other NLU tasks, this paper studies the effectiveness of OpenAI GPT-based models, including GPT-3, InstructGPT, GPT-3.5, and GPT-4 for sarcasm detection. Authors have considered twelve variants of these models, including all in total for the study, and have considered a balanced SARC (sarcastic Reddit comments) “pol-bal” dataset to validate the effectiveness of the different models. They have fine-tuned and tested the base GPT-3 models and zero-shot tested the GPT-3, InstructGPT, GPT-3.5, and GPT-4 models over the pol-bal dataset with and without domain context prompt, and zero-shot prompt. The study shows that a fine-tuned GPT-3 davinci model with and without domain context achieves state-of-the-art results (accuracy = 0.81 and F 1 = 0.81). Towards zero-shot, GPT-4 GPT-4-0613 reported an accuracy ≈ 0.71 and F 1 ≈ 0.75, which seems a good performance; however, it is unable to beat the prior models and stands second in the list. Including domain context in fine-tuning and zero-shot learning cases seemed mostly irrelevant and could not help the models improve. Although the latest version of GPT-4 performed better than the earlier one, based on the observation that the GPT-3.5-turbo model’s performance has declined from release to release, the GPT model’s ability to detect sarcasm may decline or improve with new releases.

**Strengths:**

1. The proposed study leverages the potential of OpenAI GPT models for sarcasm detection, an important NLU problem.
2. The results obtained over the fine-tuned models are promising; specifically, the fine-tuned GPT-3 davinci model outperforms the prior sarcasm detection models.
3. Sarcasm is understood between people or groups based on shared knowledge; in that case, the notable performance of the studied models is laudable but also gives the assumption that the respective models might have seen the pol-bal dataset during training.

**Weaknesses:**

1. The study is validated over just one balanced dataset; however, in the real world, sarcastic and non-sarcastic conversations rarely occur equally likely over any social media platform.
2. This study can not be generalized to other datasets.
3. The prompt with domain context is trivial, and the reported result shows that it is not helping the models towards sarcasm detection.
4. A detailed study is missing regarding the possible reason behind the model outperforming the prior models in a case.
5. In many cases, the models are performing even inferior to ZeroR classifier.
6. Based on the results reported in Table 3, it seems most of the models' performance is inferior to ZeroR classifier in the case of zero-shot.

**Questions:**

1. Why have the authors not considered any other LLMs?
2. Is there no scope for feature engineering in this approach for sarcasm detection?

---

### Author Response · Authors · 2023-11-21

Our sincere thanks go out to the reviewers for their valuable comments and suggestions. We will address them and resubmit the paper.

---

### Meta-Review · Area_Chair_Mgrj · 2023-12-06

**Metareview:**

This paper tackles the task of sarcasm detection, which is a challenging problem in NLP. The authors address this by investigating the performance of a number of OpenAI GPT-based models. The paper has very little novelty as it largely focuses on the effectiveness of existing methods. Furthermore, a key weakness is the exclusive focus on models that are not open-source, limiting its contributions and usefulness to the research community. The reviewers have raised a number of concerns; however, the authors have not really participated in the rebuttal. I agree with the reviewers that a more interesting research question is that of the effect of domain information.

**Justification For Why Not Higher Score:**

Very limited novelty as the paper only focuses on investigating the effectiveness of existing GPT-based methods.

**Justification For Why Not Lower Score:**

NA

---

### Decision · Program_Chairs · 2024-01-16

Reject